# Cerebrospinal fluid oligoclonal bands in Neuroborreliosis are specific for Borrelia burgdorferi

**Klaus Berek**[ID][☯], **Harald Hegen**[☯], **Michael Auer, Anne Zinganell, Franziska Di Pauli, Florian Deisenhammer**\*

Department of Neurology, Neuroimmunology Laboratory, Medical University of Innsbruck, Innsbruck, Austria

☯ These authors contributed equally to this work.
\* florian.deisenhammer@tirol-kliniken.at

**Data Availability Statement:** All relevant data are within the manuscript and its Supporting Information files.

## Abstract

### Background

Cerebrospinal fluid (CSF) oligoclonal bands (OCB) occur in chronic or post-acute phase of inflammatory diseases of the central nervous system.

### Objective

To determine whether CSF OCB in patients with neuroborreliosis (NB) are specific for borrelia burgdorferi senso lato.

### Methods

We performed isoelectric focusing followed by immunoblotting in CSF of 10 NB patients and 11 controls (7 patients with multiple sclerosis, 2 patients with neuromyelitis optica spectrum disease, 1 patient with dementia and 1 patient with monoclonal gammopathy). Immunoblotting was performed using an uncoated as well as a borrelia antigen pre-coated nitrocellulose membrane (NCM). OCB were counted by visual inspection and photometric analysis. OCB were compared between uncoated und pre-coated NCM both in the NB and control group. For validation purposes inter-assay precision was determined by calculating the coefficient of variation (CV)

### Results

Borrelia-specific OCB were found in the CSF of 9 NB patients and in none of the control subjects resulting in a sensitivity of 90% and a specificity of 100%. Number of NB specific OCB were 11±7 bands by photometric analyses compared to 9±5 bands by visual inspection. Validation experiments revealed an inconsistent inter-assay precision between visual and photometric analyses (NB uncoated: visual 28% versus photometric 14%, control subject uncoated: visual 16% versus photometric 24%).

**Funding:** Borrelia antigens were provided free of charge by Euroimmun. URL: https://www.euroimmun.com/startseite.html The funders had no role in study design, data collection and analysis, decision to publish, or preparation of the manuscript.

**Competing interests:** The authors have read the journal's policy and the authors of this manuscript have the following competing interests: KB has participated in meetings sponsored by and received travel funding from Roche. HH has participated in meetings sponsored by, received speaker honoraria or travel funding from Bayer, Biogen, Merck, Novartis, Sanofi-Genzyme, Siemens, Teva, and received honoraria for acting as consultant for Teva and Biogen. MA received speaker honoraria and/or travel grants from Biogen, Merck, Novartis and Sanofi. AZ has participated in meetings sponsored by, received speaking honoraria or travel funding from Biogen, Merck, Sanofi-Genzyme and Teva. FDP has participated in meetings sponsored by, received honoraria (lectures, advisory boards, consultations) or travel funding from Bayer, Biogen, Merck, Novartis, Sanofi-Genzyme, Teva, Celgene and Roche. FD has participated in meetings sponsored by or received honoraria for acting as an advisor/speaker for Almirall, Alexion, Biogen, Celgene, Genzyme-Sanofi, Merck, Novartis Pharma, Roche, and TEVA ratiopharm. His institution has received research grants from Biogen and Genzyme Sanofi. He is section editor of the MSARD Journal (Multiple Sclerosis and Related Disorders). There are no patents, products in development or marketed products to declare. This does not alter our adherence to PLOS ONE policies on sharing data and materials.

## Conclusions

In CSF samples with positive OCB, Borrelia-specific bands were detected in almost all NB patients and in none of the control subjects. Inconsistent inter-assay precision may be explained by a poor comparability of visual and photometric approach.

## Introduction

Neuroborreliosis (NB) is a tick-borne infection of the nervous system that is widespread in Europe and North America and caused by the spirochetes Borrelia burgdorferi sensu stricto, Borrelia garinii and Borrelia afzelii [1–3]. Approximately 10–15% of all borrelia infections affect the nervous system, most commonly leading to meningitis, cranial (poly-)neuritis or (poly-)radiculitis [1, 4]. Besides the typical neurological symptoms, diagnosis of NB requires CSF pleocytosis and intrathecally produced anti-borrelia specific antibodies [5]. Other CSF abnormalities include elevated CSF protein levels as well as positive oligoclonal bands (OCB) [1, 6].

Oligoclonal bands as detected by isoelectric focusing (IEF) and subsequent immunoblotting are the gold standard to generally assess an intrathecal immunoglobulin (Ig) G synthesis. OCB are an indicator of chronic or post-acute inflammation of the central nervous system (CNS) or the spinal nerve roots, both compartments that are typically affected in NB. However, OCB occur also in a variety of other inflammatory neurological diseases, particularly in multiple sclerosis (MS) where OCB are part of the diagnostic criteria [7–11].

The proof that OCB are specific to certain antigens in different neurological diseases has been a challenging goal in the last decades. A few studies showed that OCB are directed against CV2/CRMP5, amphiphysin, HuD, Yo [12–14] and Ri [15] in patients suffering from paraneoplastic neurological syndromes as well as in infectious diseases against the Human Immunodeficiency Virus (HIV) [16] and borrelia [17]. In these studies, OCB were detected on blots that had been pre-coated with the respective antigens [12–17]. However, studies on NB patients are rare and there are some methodological limitations, e.g. inappropriate control subjects as well as assay specificity issues [17].

Our study was designed to determine sensitivity and specificity of borrelia burgdorferi senso lato specific OCB in patients with NB.

## Methods

In this retrospective, cross-sectional case-control study we included 10 patients with NB and 11 controls. NB was diagnosed according to the EFNS criteria [5], i.e. all patients had typical neurological symptoms, CSF pleocytosis, and elevated anti-borrelia specific antibody index ($>$1.5). The control group comprised 7 patients with MS, 2 patients with neuromyelitis optica spectrum disease, 1 patient with dementia and 1 patient with monoclonal gammopathy. All NB patients and 10 patients of the control group showed CSF-restricted OCB (pattern II), 1 control subject had mirror bands in CSF and serum (pattern IV) [8].

We performed IEF followed by immunoblotting using uncoated nitrocellulose membrane (NCM) as well as borrelia antigen pre-coated NCM in CSF and serum sample pairs from NB patients and controls.

### Isoelectric focusing followed by immunoblotting

IEF and immunoblotting was performed as previously published [18], using (A) uncoated NCM and (B) borrelia antigen pre-coated NCM.

(A) 15 μl of CSF and serum (diluted in Aqua dest. to achieve an IgG concentration of 3 mg/dl) were applied to polyacrylamide gel (7.5%) covering a pH range of 3–10. Isoelectric focusing was carried out using 1N $H_3PO_4$ for the anode and 1N NaOH for the cathode. The samples were run for 2 hours (1.08 kV, 15 mA, 200W). After that, gels were mechanically blotted on NCM over 20 minutes. Membranes were then placed in blocking solution (20 g/L dried, skimmed milk in 0.9% NaCl) for 30 minutes and washed three times with 0.9% NaCl. For immunolabelling, membranes were incubated for 1 hour with goat anti-human IgG (Cat. No. 2040–01, Southern Biotech, Birmingham, AL, USA) diluted 1:2000 in 50 ml diluent (2 g/L dried, skimmed milk in 0.9% NaCl). Rinsing with tap water for ten times was followed by one wash in diluent for 5 minutes. Thereafter, membranes were incubated with horseradish peroxidase-labelled rabbit anti-goat IgG (Ca. No. P0160, Agilent Dako, Santa Clara, CA, USA) diluted 1:1000 in 50 ml diluent for 1 hour. Another rinsing with tap water for ten times was followed by one wash in 0.9% NaCl for 5 minutes. Staining was performed by using 25 mg of 3-amino-9-ethylcarbazole diluted in 10 ml ethanol and 50 ml acetate buffer. After adding 50 μl of 30% hydrogen peroxide, membranes were incubated for 15 minutes. After development of the red-brown bands, membranes were washed with distilled water and air-dried.

(B) IEF and immunoblotting were performed as above (A), apart from pre-coating the NCM. For pre-coating, we incubated the NCM with borrelia antigens dissolving a mixture of VlsE (variable major protein-like sequence expressed) antigens of B. afzelii (Euroimmun Charge-B: B081117RF), B. garinii (Euroimmun Charge-B: B081117RH), and B. burgdorferi sensu lato (Euroimmun Charge-B: B081117RK) in 30mL 0.9% NaCl. After an incubation time of one hour and three runs of washing membranes with 0.9% NaCl NCMs were used for immunoblotting.

Following the recommendations of Freedman et al. [8, 19] the membranes were visually inspected and OCB counted by three independent blinded neurologists experienced in OCB analysis. OCB positivity was defined as >2 bands in CSF without corresponding bands in serum [18]. Additionally, we determined the number of OCB by a semiquantitative method using ImageJ, a free software based on Java for the digital evaluation of e.g. Western blots [20, 21]. For that purpose, NCM were scanned by CanoScan LIDE70, edited by Photoshop CS5 (Version 5/ 2010; Adobe Inc., San José, CA, USA), transformed to a TIFF-file and, finally, a window of evaluation of 207x564 Pixel was manually selected for analysis (Fig 1). ImageJ returned segmented peaks, each corresponding to a single CSF band.

## Validation and inter-assay precision

To determine inter-assay precision, six repetitive IEF runs of CSF samples from one NB and one control patient followed by immunoblotting using uncoated and pre-coated NCM was performed. Subsequently, OCB were counted by visual inspection and by photometric analyses with ImageJ. The coefficient of variation (CV), was calculated by dividing standard deviation (SD) by the mean number of OCB.

## Statistical analysis

Statistical analysis was performed using SPSS 26.0 (SPSS Inc., Chicago, IL, USA). Frequency distributions were analysed by χ2 test. Comparisons of non-parametric data such as number of OCB between groups was performed by Mann-Whitney U test. A p-value <0.05 was considered statistically significant.

## Ethics

According to national regulations no ethical vote was needed because this is an anonymous retrospective re-analysis of existing samples that were obtained in routine diagnostic

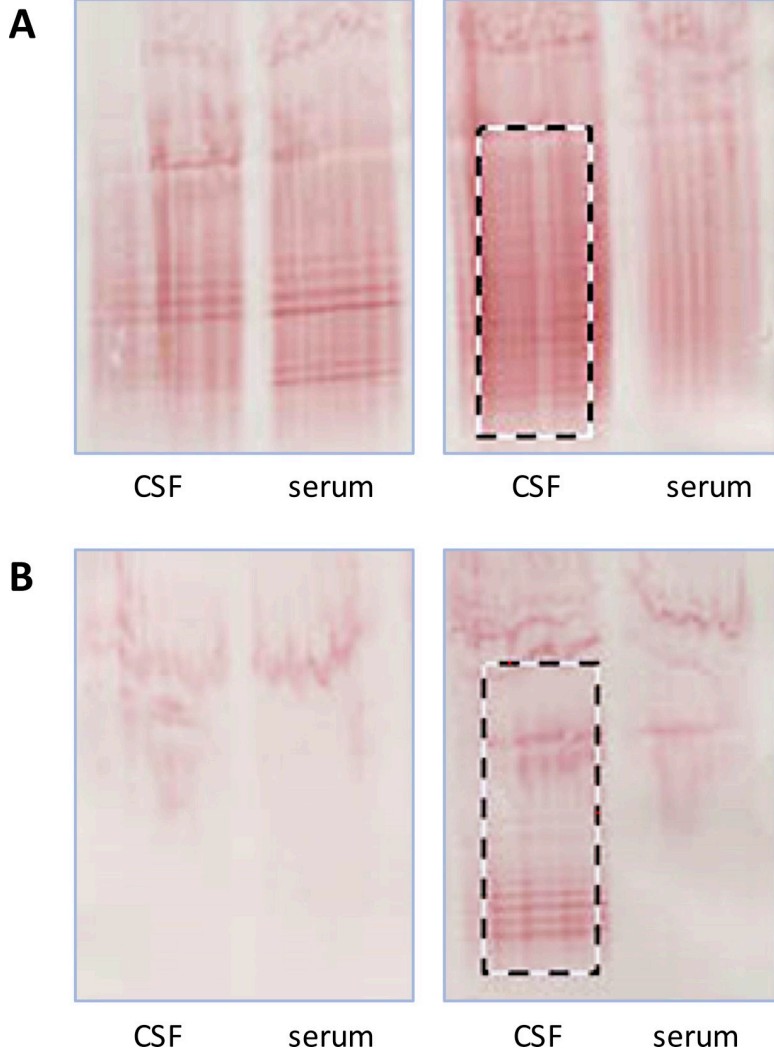

CSF          serum          CSF          serum

CSF          serum          CSF          serum

**Fig 1.** Oligoclonal bands on (A) uncoated and (B) pre-coated nitrocellulose membrane. The control patient showed mirror bands in CSF and serum (type IV) on the uncoated NCM (Fig 1A, left), while no bands could be detected using the pre-coated NCM (Fig 1B, left). The neuroborreliosis patient had OCB in the CSF but not in serum on the uncoated NCM (Fig 1A, right), and the same pattern (type II) using the pre-coated NCM (Fig 1B, right). Framed areas were used for semi-quantitative determination of oligoclonal bands by ImageJ. Abbreviations: CSF: cerebrospinal fluid. NCM: nitrocellulose membrane. OCB: oligoclonal bands

procedures. Patients signed a standard form provided by the hospital allowing use of excess material from routine diagnostic procedures. We adhered to the declaration of Helsinki and the Austrian Data Safety Authority instructions (www.ris.bka.gv.at., 2018) by anonymising all study participants' data.

## Results

### Borrelia-specific OCB

A total of 10 patients with NB and 11 controls were included into the study. Demographic characteristics and main CSF findings are displayed in Table 1.

In the NB group, 9 of 10 patients had borrelia-specific OCB, i.e. bands on uncoated as well as on pre-coated NCM. One patient had OCB on the uncoated NCM, but no bands on the

**Table 1. Demographic and CSF findings in NB patients and controls.**

|  | Neuroborreliosis | Controls |
|---|---|---|
| N | 10 | 11 |
| Age (years), mean±SD | 64±17 | 37±15 |
| Sex (female), n (%) | 5 (50) | 7 (64) |
| CSF WBC count (/μl), median (IQR) | 141 (3–401) | 20 (4–59) |
| CSF RBC count (/μl), median (IQR) | 4 (0–9) | 0 (0–10) |
| CSF/ serum glucose ratio, median (IQR) | 0.49 (0.41–0.57) | 0.57 (0.55–0.70) |
| CSF total protein (g/l), median (IQR) | 1.87 (0.57–1.98) | 0.44 (0.38–0.50) |
| $Q_{alb}$, median (IQR) | 29.9 (6.9–33.7) | 6.1 (4.3–7.7)[3] |
| IgG index, median (IQR) | 0.73 (0.65–1.14) | 1.01 (0.95–1.44)[3] |
| IgG intrathecal synthesis[1], n (%) | 10 (100) | 9 (90)[3] |
| IgG IF (%)[2], median (IQR) | 28 (5–51) | 34 (24–51)[3] |
| IgA intrathecal synthesis[1], n (%) | 1 (10) | 3 (30)[3] |
| IgA IF (%)[2], median | 24[4] | 19[3,4] |
| IgM intrathecal synthesis[1], n (%) | 7 (78)[5] | 7 (70)[3] |
| IgM IF (%)[2], median (IQR) | 73 (52–84)[5] | 34 (24–71)[3] |
| Oligoclonal bands, n (%) |  |  |
| Pattern II[6] | 10 (100) | 10 (91) |
| Pattern IV[6] | 0 | 1 (9) |
| Anti-borrelia specific antibody index, median (IQR) | 25.42 (12.67–67.97) | Not done |

[1]Intrathecal Ig synthesis as determined by Reiber hyperbolic function [22] was present when intrathecal fraction was >0%.

[2]Median (IQR) percentage intrathecal fraction was calculated using patients with present Ig synthesis (i.e. IF>0%) only.

[3]Data available of 10 patients. The patient with missing data was that with monoclonal gammopathy.

[4]Median values are presented. Due to the low number of patients with presence of intrathecal IgA synthesis IQR could not be provided.

[5]Data available of 9 patients.

[6]OCB were determined for routine diagnostic purposes, and classified into the pattern as defined by Freedman et al. [8].

*Abbreviations*: CSF: cerebrospinal fluid. IF: intrathecal fraction. Ig: immunoglobulin. $Q_{alb}$: CSF/ serum albumin quotient. RBC: red blood cells. SD: standard deviation. WBC: white blood cells. IQR: interquartile range.

pre-coated NCM resulting in a diagnostic sensitivity of 90%. The number of OCB were similar between uncoated and pre-coated NCM as determined by visual inspection: mean 12±5 vs. 9 ±5 bands (p = 0.190). On pre-coated membranes four patients (40%) showed a lower number of OCB (on average 6±4 bands less), in four patients (40%) the OCB count was identical, one patient (10%) showed a higher OCB count in pre-coated than in uncoated NCM (2 bands more), and in one patient (10%) no OCB could be recovered using pre-coated membranes.

In the control group, all patients had OCB (on average 10±5 bands) on uncoated NCM. On pre-coated NCM one patient showed 2 OCB, which was considered negative, and 10 patients had no OCB resulting in a diagnostic specificity of 100%.

Using ImageJ software OCB counts in NB patients yielded a mean (±SD) of 12±4 bands on uncoated and 11±7 bands on pre-coated NCM (p = 0.579). Six patients (60%) showed a lower number of OCB on pre-coated NCM (on average 3±1 bands less), one patient had more OCB (12 bands more) on pre-coated than on uncoated NCM (10%), in two patients (20%) the OCB count was identical and in one patient (10%) no OCB could be found on pre-coated NCM.

**Table 2. Number of OCB on uncoated and pre-coated nitrocellulose membranes among Neuroborreliosis and control patients.**

| | Visual analysis | | Photometric analysis | |
|---|---|---|---|---|
| | **Uncoated NCM** | **Pre-coated NCM** | **Uncoated NCM** | **Pre-coated NCM** |
| **Neuroborreliosis** | | | | |
| OCB positive n (%) | 10 (100) | 9 (90) | 10 (100) | 9 (90) |
| No. of OCB mean±SD, median (IQR) | 12±5 11 (8–15) | 9±5 9 (7–11) | 12±4 12 (9–13) | 11±7 10 (7–14) |
| **Controls** | | | | |
| OCB positive n (%) | 11 (100) | 0 (0) | 10 (91) | 0 (0) |
| No. of OCB mean±SD, median (IQR) | 10±5 9 (6–15) | 0.2±0.6 0 (0–0) | 9±5 9 (6–15) | 0.2±0.6 0 (0–0) |

*Abbreviations*: NCM: nitrocellulose membrane. No.: number. OCB: oligoclonal bands. SD: standard deviation.

In the control group, photometric analysis detected OCB in 10 subjects (90.9%) using uncoated NCM (on average 9±5 bands). On pre-coated membranes one subject (the same as with visual inspection) showed 2 bands, while the remaining 10 subjects had no OCB at all.

Comparing the visual inspection to the photometric analysis revealed similar results for all sub-investigations: NB patients on uncoated NCM showed 12±5 bands as measured by visual inspection and 12±4 by photometric analysis (p = 1.0). On pre-coated NCM the NB group showed a mean of 9±5 bands by visual and 11±7 by photometric examination (p = 0.579).

In the control group 10±5 OCB were counted on uncoated membranes by visual and 9±5 by photometric inspection (p = 1.0). On pre-coated membranes all control subjects showed a negative OCB result both by visual and photometric analyses. 10 control patients showed no bands, one subject had a minimal number of OCB (2 bands) on pre-coated NCM both by visual and photometric evaluation (Table 2).

## Methodologic validity

One NB patient and one control subject were used for replication experiments (S1 and S2 Tables). The NB patient showed OCB both on uncoated and on pre-coated NCM in every of the six IEF runs as determined by visual inspection, with an average of 14±4 bands (using uncoated NCM) and 10±3 bands (using pre-coated NCM) with a CV of 28% and 29%. In the control patient, OCB were found on uncoated NCM in every run (average 10±2; CV 16%), whereas no bands were recovered on pre-coated NCM.

The photometric approach revealed an average of 13±2 bands (CV 14%) on uncoated and 10±3 (CV 27%) bands on pre-coated membranes. Within the control subject, every IEF run on uncoated NCM revealed OCB (average 9±2; CV 24%) and no one on pre-coated NCM.

## Discussion

We demonstrated high specificity of clonally expanded IgG against borrelia antigens not only by detection of OCB on borrelia pre-coated membranes but also by using appropriate controls, i.e. OCB derived from other entities, none of which reacted with borrelia antigens. Similar results have been obtained earlier with somewhat less sensitive methods and different antigens demonstrating persistence of borrelia specific OCB [17]. In this prior study one third of patients with borrelia-specific OCB had no total IgG oligoclonal bands, a finding that rises some questions on methodology. Other studies could demonstrate specificity of OCB of

onconeural antibodies in paraneoplastic neurological syndromes [12–15] as well as HIV antibodies [16].

To our knowledge, performance characteristics such as assay precision have not been determined in previous work. This is an important aspect, particularly because IEF followed by immunoblotting is a qualitative method. Nevertheless, quantification of OCB counts is needed at least for distinction between positive and negative results as one single band is not a proof of oligoclonal or monoclonal IgG expansion [18]. In the present study, visual inspection by experienced persons resulted in similar average OCB counts compared to computer-aided automatic counting using ImageJ software with CVs ranging from 14%-29% which is acceptable given the pseudo-quantitative nature of OCB counting. Similarly, previous studies on conventional OCB testing found that classification into positive or negative results shows a high reproducibility and low inter-rater variability, whereas counting absolute numbers of CSF bands is less reliable [23, 24].

We observed that 40% of NB patients showed a lower absolute number of OCB on precoated than on uncoated NCM and in one patient we couldn't detect OCB on precoated NCM at all. This discrepancy might be explained by a polyspecific intrathecal B cell response with some clones directed against borrelia and other clones directed against either unrelated or antigens that were not included in the VlsE mix used for coating such as Flagellin, BmpA, or OspC [25, 26]. However, the rationale to use a mixure of VlsE was that these antigens are the most significant in borrelia antibody testing, e.g. there are also widely used in enzyme-linked immunosorbent assays (ELISA) [25, 26]. Similar findings were reported in patients with demyelinating CNS diseases who frequently show OCB bands that are directed against a variety of known and unknown antigens [27].

One control patient had 2 borrelia specific bands which we considered negative because we use a cut-off >2 bands for OCB positivity. We would like to state that no certain OCB cut-off has been recommended by any CSF guideline yet, however, a number of studies used either >1 or >2 CSF-restricted bands to define OCB positivity. Current evidence is in favour of a cut-off of >2 CSF bands resulting in high specificity (of >95%, i.e. allows less than 5% false positive results which is accepted and tolerable in laboratory diagnostics) (18). There are several explanations supporting this strict cut-off. First, the assessment of OCB by visual inspection is rater-dependent, and previous studies have shown that inter-rater agreement on the number of OCB, i.e. to decide on single bands, is only low [23, 24, 28]. Furthermore, there are also methodological issues that can hamper assessing CSF bands, e.g. non-linearity of the pH gradient used for IEF [7] or physiological microheterogeneity of IgG [23]. Due to these considerations, we decided to use the same cut-off (>2 CSF bands) for evaluation of borrelia-specific bands. Besides that, from a pathophysiological point of view, it is possible that the above-mentioned patient suffering from MS might have had neuroborreliosis previously or an asymptomatic intrathecal anti-borrelia antibody response. There was not enough sample volume left to do anti-borrelia specific antibody index to examine this possibility. Another explanation might be that MS as a demyelinating CNS disease can be associated with intrathecal B cell clones that produce molecular-mimicry type antibodies that cross-react with both, self-proteins of the individual and infectious antigens. Previous studies reported OCB reactivity to HHV-6 and other infectious agents in MS patients [27].

Photometric analysis showed a slightly higher sensitivity regarding borrelia-specific bands (11±7) compared to visual inspection (9±5). Variability of the background intensity is a limiting factor as the height of the peaks depended not only on the presence of a band but was also influenced by the background colour to some extent. We addressed this issue by re-baselining the background noise and applying a standard window of evaluation. In one sample the software detected an exorbitantly higher number of bands on the precoated compared to the

uncoated membrane which turned out to be caused by a scattered background activity. This underlines the importance of visual inspection even if automated systems are used. These data are in agreement with earlier findings and highlight the importance of standardised guidelines in order to reduce potential sources of error [29].

There are some limitations of the study. The sample size was small partially due to the large amounts of antigen needed and the limited number of available samples. However, the high rate of positives in the NB and negatives in the control groups suggests a true finding. The diagnostic accuracy of borrelia-specific OCB (sensitivity 90%, specificity 100%) seems to be comparable to those of the anti-borrelia specific antibody index (sensitivity 75%, specificity 95%) [30] and CXCL13 (sensitivity and specificity according to recent studies up to approximately 95%) [31–33].Further studies with larger sample sizes would be needed to reliably confirm these findings. As this was an exploratory work validation experiments were restricted to inter-assay precision measurements. Counting bands on blots is critical due to the nature of a pseudo-quantitative method and using a new software that is originally not designed for use in the present context, however, performing similarly. Cross-reactivity with other spirochetes cannot be entirely ruled out. Although we did not run blots on membranes precoated with antigen from non-borrelia spirochetes all samples of NB patients had high titer antibodies against borrelia in the ELISA, typical routine CSF findings, and corresponding clinical syndromes typical of NB. Also, anti-borrelia antibodies cross-react less with other spirochetes than vice versa [34].

Taken together, we provide evidence that borrelia-specific OCB are present in a majority of NB patients. We feel that this method has a potential to identify the specificity of OCB in other inflammatory CNS diseases including those with still unknown cause or unidentified antigens.

## Supporting information

**S1 Table. Number of OCB as obtained by six IEF runs in one NB patient assessed by visual and photometric analysis.** Abbreviations: NCM: nitrocellulose membrane. NB: neuroborreliosis. OCB: oligoclonal bands. SD: standard deviation.
(PDF)

**S2 Table. Number of OCB as obtained by six IEF runs in one control patient assessed by visual and photometric analysis.** Abbreviations: NCM: nitrocellulose membrane. OCB: oligoclonal bands. SD: standard deviation.
(PDF)

**S1 Fig. Immunoblots of the CSF of Neuroborreliosis patients on uncoated and borrelia-antigen pre-coated membranes.**
(TIFF)

**S2 Fig. Immunoblots of the CSF of control patients on uncoated and borrelia-antigen pre-coated membranes.**
(TIFF)

**S1 Raw image.**
(PDF)

## Acknowledgments

We thank Josef Haus for laboratory assistance.

## Author Contributions

**Conceptualization:** Klaus Berek, Harald Hegen, Florian Deisenhammer.

**Data curation:** Klaus Berek, Harald Hegen.

**Formal analysis:** Klaus Berek, Harald Hegen.

**Funding acquisition:** Florian Deisenhammer.

**Investigation:** Klaus Berek, Harald Hegen.

**Methodology:** Klaus Berek, Harald Hegen, Florian Deisenhammer.

**Project administration:** Klaus Berek, Harald Hegen, Florian Deisenhammer.

**Resources:** Florian Deisenhammer.

**Software:** Klaus Berek, Harald Hegen.

**Supervision:** Harald Hegen, Florian Deisenhammer.

**Validation:** Klaus Berek, Harald Hegen.

**Visualization:** Klaus Berek.

**Writing – original draft:** Klaus Berek.

**Writing – review & editing:** Harald Hegen, Michael Auer, Anne Zinganell, Franziska Di Pauli, Florian Deisenhammer.

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
