## [Decision Letter · Decision Letter 0]

28 Apr 2020

PONE-D-20-06818

Cerebrospinal fluid oligoclonal bands in Neuroborreliosis are specific for Borrelia burgdorferi

PLOS ONE

Dear Prof. Deisenhammer,

Thank you for submitting your manuscript to PLOS ONE. After careful consideration, we feel that it has merit but does not fully meet PLOS ONE’s publication criteria as it currently stands. Therefore, we invite you to submit a revised version of the manuscript that addresses the points raised during the review process.

We would appreciate receiving your revised manuscript by Jun 12 2020 11:59PM. To enhance the reproducibility of your results, we recommend that if applicable you deposit your laboratory protocols in protocols.io, where a protocol can be assigned its own identifier (DOI) such that it can be cited independently in the future. For instructions see: http://journals.plos.org/plosone/s/submission-guidelines#loc-laboratory-protocols

We look forward to receiving your revised manuscript.

Kind regards,

Edgar Meinl, M.D.

Academic Editor

PLOS ONE

Journal Requirements:

'Borrelia antigens were provided free of charge by Euroimmun.

URL: https://www.euroimmun.com/startseite.html

The funders had no role in study design, data collection and analysis, decision to

publish, or preparation of the manuscript.'

We note that you received funding from a commercial source: [Name of Company]

Additional Editor Comments (if provided):

Reviewers' comments:

Reviewer's Responses to Questions

**Comments to the Author**

1. Is the manuscript technically sound, and do the data support the conclusions?

Reviewer #1: Yes

Reviewer #2: Yes

2. Has the statistical analysis been performed appropriately and rigorously? 

Reviewer #1: Yes

Reviewer #2: Yes

3. Have the authors made all data underlying the findings in their manuscript fully available?

Reviewer #1: Yes

Reviewer #2: Yes

4. Is the manuscript presented in an intelligible fashion and written in standard English?

Reviewer #1: Yes

Reviewer #2: Yes

5. Review Comments to the Author

Reviewer #1: The authors established a test to detect, whether CSF OCBs in patients with neuroborreliosis (NB) are specific for borrelia burgdorferi senso lato. By applying an immunoblotting test, the authors suggest that OCBs in LN are indeed specific against borrelia burgdorferi. The major limitation of this study is the limited number of patients with LN (n=10) and controls (n=11), however, the authors state in the discussion, that this is an exploratory study.

Limitations

What antigens for borrelia burgdorferi are available and what borrelia antigen has been used for pre-coated NCM (page 4, line 85)? The author could shortly refer to antigens (also in comparison to ELISA tests) and their specificity – also in regard to cross reactivities (maybe see Chandra A et al., 2011 Epitope mapping…).

The authors could shortly summarize common borrelia specific tests and their sensitivity / specificity (maybe as a table). For example CSF CXCL13 has been suggested as a clinical test to detect NB, although CXCL13 is not specific for NB (Rupprecht et al., 2018, Diagnostic value of CSF CXCL13…; Lepennetier et al, 2019, Cytokine and immune cell…).

What is the advantage of the new test?

Could the test be introduced in a clinical setting to improve diagnostics?

Why do the authors apply a cut-off of > 2 bands for the pre-coated NCM test? Although OCBs are usually considered positive when >2 bands are specifically detected in CSF, why should this approach by applied for the immunoblotting test, especially in the context of the positive control? Please reconsider this issue.

It would be interesting, if the control patient (page 9, line 181) with positive OCBs in the immunoblotting test has a positive antibody titer – at least in serum.

Table 1: CSF standard finding could be more detailed (e.g. Ig index, intrathecal Ig production, glucose. Lactate …)

Picture quality of Figure 1A right is difficult which may be due to submission format.

It would have been interesting to correlate CXCL13 values (as a measure of B cell activity) with OCBs and test results.

Reviewer #2: In this retrospective, cross-sectional case-control study CSF-serum sample pairs of 21 patients (10 with neuroborreliosis and 11 OCB positive controls) were included. The authors aimed to determine whether CSF OCBs in patients with neuroborreliosis are specific for borrelia burgdorferi. Isoelectric focusing followed by immunoblotting using (A) an uncoated as well as (B) a borrelia antigen pre-coated nitrocellulose membrane was performed. The results indicate borrelia-specific bands in 9 of 10 neuroborreliosis patients. The manuscript is well written and concise.

It might profit in view of the reviewer from additionally addressing a few minor points:

(1) Methods section: please indicate that CSF-serum sample pairs were analyzed.

(2) Figure 1 could be extended by another example C (OCB type 2, e.g., MS CSF-serum

pair).

(3) Please discuss the possible applications of this test method. Are there any

advantages with regard to the current standard (sample volume, costs, ..)?

(4) In table 2: Are the data normally distributed? If not, please indicate the median

and range or IQR. Please indicate the OCB positivity rate in each group. Visual analysis

NB uncoated 100 vs pre-coated 90%, Controls 100 vs 0%; photometric analysis NB 100 vs 90%

(?), Controls 90,9% vs

0% (?).

(5) Not sure whether the sample size is sufficient to make statements on diagnostic

accuracy, although the results are conclusive. This could be determined in a larger study?

More from academic interest than practical relevance: Is there a correlation between

anti-borrelia specific antibody indices and the number of borrelia specific OCBs?

6. PLOS authors have the option to publish the peer review history of their article (what does this mean?). If published, this will include your full peer review and any attached files.

Reviewer #1: No

Reviewer #2: No

---

## [Author Response · Author response to Decision Letter 0]

6 Aug 2020

Journal Requirements: 

Reply:

Uploaded files have been named as requested. 

Reply:

We have added a supporting information file with the images used for analyses. However, the images are somewhat cropped because it is required for the image processing software which wouldn’t work otherwise. Nevertheless, we can confirm that the scans contain the whole region of interest, i.e. the area (lanes) where sample material got blotted on membranes.

'Borrelia antigens were provided free of charge by Euroimmun.

URL: https://www.euroimmun.com/startseite.html

The funders had no role in study design, data collection and analysis, decision to

publish, or preparation of the manuscript.' 

Reply:

We have added this statement to the Financial Disclosures section. 

We note that you received funding from a commercial source: Euroimmun 

Reply:

We have explicitly stated the funder “Euroimmun” and included the sentence “This does not alter our adherence to PLOS ONE policies on sharing data and materials” in the Competing Interests Statement. We also included this amendment within the cover letter.

 

Response to the Reviewers:

Reviewer #1: The authors established a test to detect, whether CSF OCBs in patients with neuroborreliosis (NB) are specific for borrelia burgdorferi senso lato. By applying an immunoblotting test, the authors suggest that OCBs in LN are indeed specific against borrelia burgdorferi. The major limitation of this study is the limited number of patients with LN (n=10) and controls (n=11), however, the authors state in the discussion, that this is an exploratory study.

Limitations

1. What antigens for borrelia burgdorferi are available and what borrelia antigen has been used for pre-coated NCM (page 4, line 85)? The author could shortly refer to antigens (also in comparison to ELISA tests) and their specificity – also in regard to cross reactivities (maybe see Chandra A et al., 2011 Epitope mapping…).

Reply: 

Thanks for the comment. In our experiments, we used a mixture of VlsE antigens of B. afzelii (Euroimmun Charge-B: B081117RF), B. garinii (Euroimmun Charge-B: B081117RH), and B. burgdorferi sensu lato (Euroimmun Charge-B: B081117RK), as stated in the “Methods/ Isoelectric focusing followed by immunoblotting” section (page 6 line 110 – 116). To our knowledge, these are among the most relevant borrelia antigens used for immunoblots as well as for ELISA; besides that, there are also several other antigens, e.g. OspC, Flagellin, BmpA, etc.. We have added a passage to the discussion section that explains why we have chosen VlsE antigens for our experiments and addressed the issue of specificity. 

2. The authors could shortly summarize common borrelia specific tests and their sensitivity / specificity (maybe as a table). For example CSF CXCL13 has been suggested as a clinical test to detect NB, although CXCL13 is not specific for NB (Rupprecht et al., 2018, Diagnostic value of CSF CXCL13…; Lepennetier et al, 2019, Cytokine and immune cell…).

What is the advantage of the new test?

Could the test be introduced in a clinical setting to improve diagnostics?

Reply: 

We performed the present experiments to investigate whether CSF OCB are specific to borrelia antigens in patients fulfilling the diagnostic criteria of neuroborreliosis. This study was driven by a fundamental interest in CSF research. As ELISA also uses VlsE as antigen and can be easily performed and the determination of antibody specificity index (ASI) is part of the diagnostic criteria, we do not think that antigen-specific OCB detection will be introduced in a routine setting. As IEF followed by OCB is per se a labor-intensive, costly, time-consuming and rater-dependent method, the pre-coating step is another further complicating this method. 

We added a passage to the discussion section relating the diagnostic value (sensitivity and specificity) of borrelia-specific OCB to the established parameters ASI and CXCL-13. 

3. Why do the authors apply a cut-off of > 2 bands for the pre-coated NCM test? Although OCBs are usually considered positive when >2 bands are specifically detected in CSF, why should this approach by applied for the immunoblotting test, especially in the context of the positive control? Please reconsider this issue.

Reply: 

Thanks for raising this important question. For the detection of “regular” OCB, no clear cut-off defining positivity is recommended by any CSF guidelines, however, most of the studies reported a cut-off of >1 or >2 CSF-restricted bands, and applying the higher cut-off results in a higher diagnostic specificity. There are several reasons for using higher cut-offs (i.e. not >0). The assessment of OCB by visual inspection is rater-dependent, and previous studies have shown that inter-rater agreement on the number of OCB, i.e. to decide to count a single band or not, is only low. Furthermore, there are also methodological issues that can hamper assessing CSF bands, e.g. non-linearity of the pH gradient used for IEF or physiological microheterogeneity of IgG. Due to these considerations, we decided to use the same cut-off (>2 bands) also for evaluation of borrelia-specific bands. We’ve added a passage that addresses this issue to the discussion section. 

4. It would be interesting, if the control patient (page 9, line 181) with positive OCBs in the immunoblotting test has a positive antibody titer – at least in serum.

Reply: 

We fully agree. However, as this was a control, ASI was not determined for routine procedures; and after performing our experiments, not enough sample volume was left to determine borrelia antibodies. We have stated this in the discussion section.

5. Table 1: CSF standard finding could be more detailed (e.g. Ig index, intrathecal Ig production, glucose. Lactate …)

Reply: 

We have added findings of the remaining CSF routine parameters (i.e. GluR, Qalb, IgG index, intrathecal fraction of IgG, IgA and IgM, OCB) to the table.

6. Picture quality of Figure 1A right is difficult which may be due to submission format.

Reply: 

Probably yes. 

7. It would have been interesting to correlate CXCL13 values (as a measure of B cell activity) with OCBs and test results.

Reply: 

We fully agree. Correlations between known diagnostic tools (e.g. CXCL13, anti-borrelia specific antibody index – see also comment of Reviewer 2) and the number of borrelia-specific OCB would be interesting. 

We did not find a correlation between anti-borrelia specific antibody index and the number of OCB on pre-coated NCM. 

We did not measure CXCL13 (as this is not part of the routine diagnostics in our clinic). 

We thank the reviewer for the careful consideration and helpful comments!

 

Reviewer #2: In this retrospective, cross-sectional case-control study CSF-serum sample pairs of 21 patients (10 with neuroborreliosis and 11 OCB positive controls) were included. The authors aimed to determine whether CSF OCBs in patients with neuroborreliosis are specific for borrelia burgdorferi. Isoelectric focusing followed by immunoblotting using (A) an uncoated as well as (B) a borrelia antigen pre-coated nitrocellulose membrane was performed. The results indicate borrelia-specific bands in 9 of 10 neuroborreliosis patients. The manuscript is well written and concise.

It might profit in view of the reviewer from additionally addressing a few minor points:

(1) Methods section: please indicate that CSF-serum sample pairs were analyzed

Reply: 

Thanks for careful reading. We explicitly stated that IEF was performed in CSF and serum sample pairs. We further included OCB pattern (according to Freedman et al. Arch Neurol. 2005;62:865-870) in the text of the methods section as well as in Table 1 (NB group: 10 patients with CSF-restricted bands (pattern II), control group: 10 patients with pattern II, 1 patient with mirror bands in CSF and serum (pattern IV)). 

(2) Figure 1 could be extended by another example C (OCB type 2, e.g., MS CSF-serum

pair).

Reply: 

Thanks for this comment. In Figure 1, we showed the example of a NB patient, who showed reproducible OCB on pre-coated NCM (i.e. borrelia-specific OCB), as well as the example of a control patient, who did not have reproducible OCB on pre-coated NCM (i.e. without borrelia-specific OCB). We feel that another example does not add further information. 

(3) Please discuss the possible applications of this test method. Are there any

advantages with regard to the current standard (sample volume, costs, ..)?

Reply: 

We performed the present experiments to investigate whether CSF OCB are specific to borrelia antigens in patients fulfilling the diagnostic criteria of neuroborreliosis. This study was driven by a fundamental interest in CSF research. As ELISA also uses VlsE as antigen, can be easily performed, as the determination of antibody specificity index (ASI) is part of the diagnostic criteria, and as IEF followed by OCB is per se a labor-intensive, costly, time-consuming and rater-dependent method and as the pre-coating step is another further complicating this method, we do not think that antigen-specific OCB detection will be introduced in a routine setting. 

(4) In table 2: Are the data normally distributed? If not, please indicate the median

and range or IQR. Please indicate the OCB positivity rate in each group. Visual analysis

NB uncoated 100 vs pre-coated 90%, Controls 100 vs 0%; photometric analysis NB 100 vs 90% (?), Controls 90,9% vs 0% (?).

Reply: 

Thanks for the comment. We have further provided median (IQR) of the OCB number as well as the n (%) of OCB positives per group.

(5) Not sure whether the sample size is sufficient to make statements on diagnostic

accuracy, although the results are conclusive. This could be determined in a larger study?

Reply: 

We fully agree. Even though the present results suggest a high diagnostic sensitivity and specificity, further studies including a larger sample size are needed to replicate these findings. We addressed this point in the discussion section. 

(6) More from academic interest than practical relevance: Is there a correlation between

anti-borrelia specific antibody indices and the number of borrelia specific OCBs? 

Reply: 

Thanks for raising this interesting point. We did not find any correlation between anti-borrelia specific antibody index and the number of borrelia-specific OCB. 

We thank the reviewer for the careful consideration and helpful comments!

---

## [Decision Letter · Decision Letter 1]

7 Sep 2020

Cerebrospinal fluid oligoclonal bands in Neuroborreliosis are specific for Borrelia burgdorferi

PONE-D-20-06818R1

Dear Dr. Deisenhammer,

We’re pleased to inform you that your manuscript has been judged scientifically suitable for publication and will be formally accepted for publication once it meets all outstanding technical requirements.

Kind regards,

Edgar Meinl, M.D.

Academic Editor

PLOS ONE

Additional Editor Comments (optional):

Reviewers' comments:

Reviewer's Responses to Questions

**Comments to the Author**

1. If the authors have adequately addressed your comments raised in a previous round of review and you feel that this manuscript is now acceptable for publication, you may indicate that here to bypass the “Comments to the Author” section, enter your conflict of interest statement in the “Confidential to Editor” section, and submit your "Accept" recommendation.

Reviewer #1: All comments have been addressed

Reviewer #2: All comments have been addressed

2. Is the manuscript technically sound, and do the data support the conclusions?

Reviewer #1: Yes

Reviewer #2: Yes

3. Has the statistical analysis been performed appropriately and rigorously? 

Reviewer #1: Yes

Reviewer #2: Yes

4. Have the authors made all data underlying the findings in their manuscript fully available?

Reviewer #1: Yes

Reviewer #2: Yes

5. Is the manuscript presented in an intelligible fashion and written in standard English?

Reviewer #1: Yes

Reviewer #2: Yes

6. Review Comments to the Author

Reviewer #1: All comments have been addressed sufficiently, and there are no further questions. The manuscript provides an interesting view on OCBs in Lyme neuroborreliosis.

Reviewer #2: I thank the authors for responding to my prior suggestions.

All comments have been addressed.

7. PLOS authors have the option to publish the peer review history of their article (what does this mean?). If published, this will include your full peer review and any attached files.

Reviewer #1: No

Reviewer #2: No

---

## [Editor Report · Acceptance letter]

15 Sep 2020

PONE-D-20-06818R1 

Cerebrospinal fluid oligoclonal bands in Neuroborreliosis are specific for Borrelia burgdorferi 

Dear Dr. Deisenhammer:

I'm pleased to inform you that your manuscript has been deemed suitable for publication in PLOS ONE. Congratulations! Your manuscript is now with our production department. 

Kind regards, 

on behalf of

Prof Edgar Meinl 

Academic Editor

PLOS ONE